# Roles of Notch Signaling in the Tumor Microenvironment

**DOI:** 10.3390/ijms23116241

**Published:** 2022-06-02

**Authors:** Antonino B. D’Assoro, Roberto Leon-Ferre, Eike-Benjamin Braune, Urban Lendahl

**Affiliations:** 1Mayo Clinic Cancer Center, Rochester, MN 55905, USA; leonferre.roberto@mayo.edu; 2Department of Cell and Molecular Biology, Karolinska Institutet, SE-17446 Stockholm, Sweden; eike-benjamin.braune@ki.se

**Keywords:** Notch signaling, cancer, tumor, tumor microenvironment, oncogene, tumor suppressor gene

## Abstract

The Notch signaling pathway is an architecturally simple signaling mechanism, well known for its role in cell fate regulation during organ development and in tissue homeostasis. In keeping with its importance for normal development, dysregulation of Notch signaling is increasingly associated with different types of tumors, and proteins in the Notch signaling pathway can act as oncogenes or tumor suppressors, depending on the cellular context and tumor type. In addition to a role as a driver of tumor initiation and progression in the tumor cells carrying oncogenic mutations, it is an emerging realization that Notch signaling also plays a role in non-mutated cells in the tumor microenvironment. In this review, we discuss how aberrant Notch signaling can affect three types of cells in the tumor stroma—cancer-associated fibroblasts, immune cells and vascular cells—and how this influences their interactions with the tumor cells. Insights into the roles of Notch in cells of the tumor environment and the impact on tumor-stroma interactions will lead to a deeper understanding of Notch signaling in cancer and inspire new strategies for Notch-based tumor therapy.

## 1. Introduction

The discovery of Notch signaling dates back to the early 1900s. The two *Drosophila* geneticists John S. Dexter and Thomas Hunt Morgan identified flies with specific wing phenotypes and characterized alleles associated with these phenotypes. In some of the mutants, the wings had indentations, notches, which later turned out to be the consequence of mutations in the Notch receptor and the notched phenotype provided the name for the Notch pathway. In a similar vein, other mutations causing wing phenotypes, such as *Delta* and *Serrate*, similarly turned out to reside in genes encoding ligands belonging to the Notch pathway. With the advent of recombinant DNA technologies, the gene encoding the *Drosophila* Notch receptor was cloned in the 1980s [1,2]. Research in *Caenorhabditis elegans* and mammals revealed that the core components of the Notch signaling are highly conserved, and we now know that Notch signaling operates in most, if not all, multicellular organisms [3,4]. The core signaling mechanism, relaying the signal from ligands on the extracellular side to gene expression changes in the nucleus, has an architecturally simple design, but despite this, it can generate very versatile downstream gene expression outputs, depending on the cell type [5] (Figure 1). Another important feature of Notch signaling is that it is highly dosage-sensitive: both hyper- and hypo-activated Notch signaling result in phenotypes in many species, an observation that is important for the role of Notch signaling in cancer, and that partly explains why mutations in genes in the Notch pathway can act as oncogenes or tumor suppressors, depending on the cellular context [6].

In this review, we will first briefly describe the molecular basis of the Notch pathway, and how the signal is transduced. Next, we will discuss the role of aberrant Notch signaling in cancer, when caused by both direct mutations in genes in the Notch pathway or dysregulated signaling output. This is followed by a section on the roles of Notch signaling in the tumor microenvironment, with a focus on cancer-associated fibroblasts, immune cells and vascular cells. 

### The Notch Signaling Pathway

Notch signaling is a cell–cell communication mechanism relaying on the interaction between membrane-spanning receptors and ligands expressed on juxtaposed cells to initiate signaling. The different steps in the Notch pathway have been extensively reviewed [3,4,7] and will therefore only be briefly summarized here. Notch receptors and ligands are structurally related in the sense that both contain a large number of (epidermal growth factor (EGF)-like repeats in their extracellular domains (Figure 1). In mammals, there are four Notch receptors (Notch1-4) and five ligands (Delta-like (Dll) 1, 3 and 4; Jagged1 and 2). The Notch receptor is a single-span transmembrane protein produced in the endoplasmic reticulum (ER) and first transported from the ER to the Golgi compartment. In the Golgi compartment, it undergoes the first of a series of proteolytic processes, the so-called site 1 (S1)-cleavage, which is carried out by furin-like convertase to produce an extracellular domain (Notch ECD) and a membrane-spanning part called transmembrane and intracellular domain (TMIC) (Figure 1). After being transported to the cell surface, the receptor interacts with a ligand located in the plasma membrane of a juxtaposed cell. The interaction between ligand and receptor involves EGF-repeat 11 and 12 in the receptor and the DSL (Delta/Serrate/Lag-2) domain in the ligand and has been characterized by X-ray crystallography [8]. The ligand–receptor interaction generates a pulling force on the receptor, which results in a conformation change in the so-called negative regulatory region (NRR), which resides at the extracellular side between the EGF-repeats and the plasma membrane [9] (Figure 1). The pulling force exerted by the ligand flips open a hinge region in the NRR, exposing a cleavage site, at which the TMIC moiety of the Notch receptor can be proteolytically processed by ADAM10 (S2-cleavage) [9]. Following S2-cleavage, which represents the ligand-regulated processing step, the membrane-spanning portion of the receptor, called Notch extracellular truncation (NEXT), is cleaved in the plasma membrane by the γ-secretase complex (S3-cleavage). In addition to Notch, the γ-secretase complex cleaves several other membrane-spanning proteins, including Amyloid Precursor Protein (APP) [10] (Figure 1). S3-cleavage liberates the C-terminal portion of the Notch receptor, called the Notch intracellular domain (Notch ICD), which relocates to the cell nucleus via the endosomal route. In the nucleus, Notch ICD binds to a DNA-binding protein called CSL (for CBF1, Suppressor of Hairless, Lag1; also known as RBPJ) and, by doing so, converts CSL from a transcriptional repressor to an activator [3,9]. The Notch ICD-CSL interaction is stabilized by Mastermind-like (Maml), and the ternary Notch ICD/MAML/CSL complex activates expression of downstream genes. The set of activated genes differ extensively between different cell types, indicating that there is only a small Notch core transcriptome that is common across cell types and thus there is considerable cell context-dependent diversity in the downstream signaling output. 

As discussed above, Notch signaling is dosage-sensitive, and to understand how Notch signaling is quantitatively tuned has been a longstanding question. Notch signaling lacks known kinase-based amplification steps in the core signaling pathway, which contrasts with, for example, Ras/MAPK and PI3K signaling, which can ramp up the signal through various kinase amplifications in the core pathway. The level of Notch signaling can, however, be tuned by posttranslational modifications of Notch ICD, which affects its half-life and, thus, Notch signaling strength. Notch ICD modifications include phosphorylation, hydroxylation, methylation, sumoylation and acetylation [11]. Among the kinases phosphorylating Notch are Nemo-like kinases, AKT, CDKs, aPKC, GSK3β and PIM kinases [12,13,14,15,16,17]. Phosphorylation by CDK8 in the PEST domain located in the C-terminal part of Notch ICD converts the PEST domain to a phosphodegron [18], which becomes ubiquitylated by the FBXW7 E3 ubiquitin ligase, inducing rapid proteasome-mediated degradation of Notch ICD [19,20,21,22]. Phosphorylation by PIM kinases is interesting because it is paralog-specific, i.e., the PIM phosphorylation sites differ between the Notch1 and Notch3 receptors, and PIM-mediated phosphorylation leads to distinct functional consequences for the two receptor paralogs [16,17]. One phosphatase, Eya1, has been identified for Notch ICD, and Eya1-induced removal of a phosphate group from a threonine residue in the transcriptional activation domain of Notch1 ICD increased the stability of Notch1 ICD [23]. Notch ICD has been shown to be hydroxylated by FIH1 [24], methylated by CARM1 [25], and acetylated by the transcriptional regulators p300 and PCAF [26,27]. Notch ICD can also be sumoylated, although the precise Notch ICD sumoylation site is not yet established [28]. 

## 2. Dysregulated Notch Signaling in Cancer

### 2.1. Oncogenic Functions of Notch

Over the last few years, it has become increasingly apparent that mutations in genes in the Notch pathway are linked to several tumor types (Figure 2). In keeping with this, Notch signaling has also moved up the ranks in the Hallmark of Cancer reviews from a less prominent position in the 2011 version [29] to be considered as one of the “canonical oncogenic drivers” in the most recent edition [30]. The first report implicating Notch mutations in cancer dates back to the observation in the early 1990s of translocations in the *NOTCH1* locus identified in patients with acute lymphoblastic T-cell leukemia (T-ALL) [31]. Soon after, it was noted that viral integrations of mouse mammary tumor virus into the mouse *Notch4* locus led to the development of mammary tumors as a consequence of the viral promoter driving expression of a truncated Notch4 ICD-like moiety [32]. Later, it was observed that, in fact, more than 50% of T-ALL patients carry activating *NOTCH1* mutations [33]. Notably, the T-ALL *NOTCH1* (and a few *NOTCH3* [34]) mutations were confined to two specific regions in the NOTCH ICD: the extracellular NRR region and the PEST domain. The mutations in the NRR region make the receptor more prone to S2-cleavage, thus reducing the requirement for ligand activation, while mutations in the PEST domain do not affect ligand activation but do prolong the longevity of the cleaved NOTCH1 ICD [35]. In support of the importance of the PEST domain and its ubiquitylation, T-ALL patients carrying loss of function mutations in *FBXW7* (the E3 ubiquitin ligase ubiquitylating the PEST domain, see above) have also been identified. Loss of *FBXW7* leads to enhanced Notch signaling by prolonging NOTCH ICD half-life by abrogation of ubiquitylation and proteasome-mediated destruction of the NOTCH ICD (see above) [22,36]. 

The activation of NRR and PEST domain mutations is not confined to T-ALL; they have also been observed in a number of other tumor types, including adenoid cystic carcinoma, while PEST domain mutations are found in chronic lymphocytic leukemia, splenic marginal zone lymphoma and diffuse large B cell lymphoma [35] (Figure 2). Furthermore, approximately 5–10% of the most aggressive form of breast cancer, triple negative breast cancer (TNBC), is associated with NRR or PEST domain mutations in *NOTCH1* and *NOTCH2* [37,38], but TNBC is also associated with elevated expression of JAGGED1, NOTCH1, NOTCH3 or NOTCH4, which in turn is associated with poor clinical prognosis [39,40,41,42]; for review see [43]. TNBC patient-derived organoids show hyperactivated Notch signaling in comparison to control organoids [44]. In other forms of breast cancer, such as estrogen receptor (ER)- or HER2-positive breast cancer, Notch signaling may initially be lower, but increases in response to endocrine therapies aiming at reducing ER activity or to therapy using HER2-blocking antibodies (trastuzumab) [45,46,47,48,49,50]; for review see [51]. The molecular basis for how ER and HER2 blockade leads to Notch activation is only partially understood and will require further research, but the notion that Notch signaling is secondarily upregulated following the primary therapies for ER- and HER2-positive breast cancer may suggest that combination therapies, involving Notch inhibition, may be an interesting future avenue to explore. A different type of NRR mutation, which generates a NOTCH3 receptor that is hyperactive but in fact never reaches the cell surface, has been identified in a family with infantile myofibromatosis [52]. How Notch signaling becomes dysregulated in the absence of mutations in the Notch pathway remains poorly understood, but it is apparent that various stress factors may enhance Notch signaling. Notably, tribbles homolog 3 (TRB3) is a stress sensor, which can activate Notch signaling in breast cancer, linked to USP9X [53,54]. Hypoxia, which is often observed in tumors, is a stress trigger that activates Notch [24,55,56], and may contribute to elevated Notch signaling in certain tumor types. 

While the link between activating Notch pathway mutations and cancer is clear, less is understood about how altered Notch signaling precisely affects tumor initiation and propagation, but Notch may affect both cancer stem cells, tumor cell metabolism, a nuclear reprogramming process called epithelial-to-mesenchymal transition (EMT) and the seeding of metastases. Notch signaling can regulate cancer stem cells in a variety of tumors, for example in breast, colon and brain tumors, which is also of relevance because cancer stem cells are linked to therapy resistance. Notch signaling has been implicated in metabolic control in glioblastoma cancer stem cells [57]. In addition, CD133^+^ glioma cancer stem cells are sensitive to γ-secretase inhibition or knockdown of *Notch* or *CSL*, indicating that Notch signaling promotes the cancer stem cell state in gliomas [58,59,60]. Furthermore, Notch contributes to stemness in non-small cell lung carcinoma [61], and inducible nitric oxide synthase (iNOS) regulates cancer stem cells in hepatocellular carcinoma via Notch signaling [62]. Notch signaling has also been implicated in resetting cellular metabolism in tumors and tuning the balance between oxidative phosphorylation and glycolysis [63]. Unconventional Notch regulates mitochondrial electron transport in cancer stem cells [64]. Furthermore, Notch is a central regulator of EMT, which is pivotal for driving tumor progression [65]. Notably, Notch signaling regulates EMT in breast and ovarian cancer, and plays a role in hypoxia-induced EMT in tumor cell lines [66]. In head and neck squamous cell carcinoma, Notch signaling is similarly implicated in enhancing EMT-mediated cancer plasticity [67], while in esophageal squamous cell carcinoma, NOTCH3 conversely reduces EMT, with a blockade of Notch promoting expression of mesenchymal markers in esophageal squamous cell carcinoma cell lines [68]. Collectively, this suggests that Notch signaling can promote or inhibit EMT-mediated cancer plasticity, depending on cellular context. The ability to seed metastasizing cells from the original tumor also involves Notch signaling. Notch signaling can promote bone metastasis of breast cancer. This process involves IL-6 [69] and blockade of Notch3 reduces bone metastasis [70]. 

There are currently no functional Notch-inhibiting therapies in clinical use [71,72], but it should be noted that there are close to 400 clinical trials listed (clinicaltrials.org). Although γ-secretase inhibitors (GSI) in principle are attractive candidates—as they block all Notch signaling—they have several side effects, which has stopped them from being used in the clinic [73]. There are however different subtypes of the γ-secretase complex, and pharmacological inhibition of only the presenilin1-containing complexes, which are most prevalent in T-ALL cells, show promising results in mouse models [74]. Another Notch therapy development approach is to generate Notch receptor or ligand paralog-specific antibodies that either block ligand-receptor interaction or hinder S2-cleavage by locking the hinge region in the NRR in a closed position [75,76]. Antibody-based approaches have yielded encouraging results in mouse model and cell line-based research, which serves as important platforms for clinical trials (see above). There are also efforts to generate small molecule inhibitors that interfere with the ternary Notch ICD/MAML/CSL complex, for example CB-103 and NADI-351 [77,78], and stapled MAML peptides, which likely break up the ternary complex, are also considered [79]. Developing therapies aiming at targeting Notch receptor posttranslational modifications, such as glycosylation of the extracellular domain, is being explored in preclinical studies [80]. 

### 2.2. Tumor Suppressor Functions of Notch

In several epithelial forms of cancer, Notch signaling acts as a tumor suppressor rather than an oncogene, i.e., loss-of-function mutations in Notch pathway genes are observed (Figure 2). This is particularly evident in various forms of squamous cell carcinomas, where, for example, 70–80% of skin squamous cell carcinoma tumors exhibit loss-of-function *NOTCH1* or *NOTCH2* mutations [81]. Loss-of-function mutations in *NOTCH1*, *NOTCH2* and *NOTCH3* are also found in squamous cell carcinoma of the lung, esophagus as well as head and neck (see [35] for review). In bladder cancer patients, activity-reducing *NOTCH1*, *NOTCH2* and *NOTCH3* mutations are likewise found, along with mutations in *MAML1* [82]. In small cell lung cancer (SCLC), inactivating mutations in *NOTCH1-4* are observed in a quarter of all SCLC patients [83]. Furthermore, *NOTCH1* mutations are more frequently observed in bronchial epithelial cells of tobacco smokers [84]. There are, however, also tumor types which are more ambiguous with regard to Notch acting as a tumor suppressor or an oncogene. Notch dysregulation in head and neck squamous cellular carcinoma represents one such case, and it is in fact possible that Notch acts both as an oncogene and tumor suppressor in a stage-dependent manner [85]. Similarly, in contrast to a role for Notch in promoting glioma cancer stem cells (see above), loss-of-function *NOTCH1* mutations have been identified in low-grade gliomas [86,87,88]. A complex role for Notch is also noted in small cell lung cancer where current data indicate that Notch signaling can be both oncogenic and tumor suppressing in the same tumor, while Notch signaling overall is growth-promoting [89]. 

The notion that Notch serves as a tumor suppressor in various types of skin or epithelial tumors while functioning as an oncogene in other tumor types may at first appear paradoxical, but is in fact in keeping with the role of Notch signaling in the respective organs. Generally, in situations where Notch acts as an oncogene, the normal role for Notch signaling during development is to serve as a gate keeper against differentiation, i.e., promoting an undifferentiated cell fate and blocking differentiation. In contrast, in the skin, where Notch is a tumor suppressor, Notch has a differentiation-promoting role [90]. The molecular differences for these opposite roles are not completely understood and are an area of active research. In its stem/progenitor cell-promoting role, Notch engages with the cell cycle machinery, for example by upregulating cyclin D3, CDK4 and CDK6 in T-cells [91]. Conversely, during skin cell differentiation, Notch instead acts as a differentiation-promoting factor that upregulates p21, thus slowing the cell cycle [92], but more research is required to fully understand the duality of Notch with regard to cancer. 

## 3. The Role of Notch Signaling in the Tumor Microenvironment

Tumor progression is not driven only by the cells carrying specific cancer-causing genetic alterations, but is also a result of activities of and interactions between the mutated cells with cells that are not mutated but infiltrate the tumor and thus constitute the tumor microenvironment or tumor stroma. Evidence is accumulating for a complex cross-talk between the tumor cells and the surrounding tumor microenvironment and decoding the mechanisms underpinning this cross-talk is a burgeoning research field [29,93]. The communication between tumors and their microenvironment can be manifested, for example, by growth-promoting factors secreted from cells in the stroma, for example, from cancer-associated fibroblasts as well as from various types of immune cells, such as macrophages- and conversely by factors released from the tumor cells that, in turn, activate macrophages in the tumor microenvironment, such as IL-4. Tumors have developed abilities to recruit immune cells from the environment into the stroma, and to convert fibroblasts from their normal relatively quiescent state to become cancer-associated fibroblasts. Furthermore, tumors reshape the normal vasculature into a tumor vasculature in order to improve access to nutrients and oxygen for a rapidly growing tumor. In the following sections, we will discuss roles played by Notch signaling in the interaction between the tumor proper and immune cells, cancer-associated fibroblasts and the tumor vasculature (Figure 3). 

### 3.1. Notch and Cancer-Associated Immune Cells

Tumors have the ability to attract immune cells and the infiltrating immune cells represent an integral component of the tumor stroma. Immune cells that antagonize as well as promote tumor growth are enrolled, but tumors have developed means to abrogate potentially tumor-antagonizing cells from attacking the tumor, and the underlying mechanisms for blocking immune surveillance are currently subject to intense analysis [29]. There is accumulating evidence that Notch signaling plays a role in tuning the immune responses in tumors, which is in line with Notch signaling playing a pivotal role in normal hematopoiesis and in generation of various types of immune effector cells [94]. Notably, Notch signaling is a key regulator of the choice between the B- and T-cell lineages and subsequently regulates the specification of various T-cell subclasses [95]. Notch signaling also plays a role in inflammatory processes, for example by exacerbating the T-cell response in the experimental autoimmune encephalomyelitis (EAE) mouse model for multiple sclerosis [96,97,98]. Furthermore, modulation of Notch regulates the severity of graft-versus-host reactions [99,100]. Modulation of T-cell responses also extends to experimental models for asthma [101,102], and recently, a role for Notch3 in regulating perivascular macrophages has been revealed [103].

There are several examples of interactions between Notch signaling and the immune response in tumors (Figure 3). Notch signaling has been demonstrated to regulate macrophage maturation towards the tumor-associated macrophage (TAM) phenotype, also referred to as M2 macrophages [104]. Activation of Notch signaling in myeloid cells decreases macrophage infiltration in pancreatic ductal adenocarcinoma [105]. Furthermore, Notch signaling can regulate inflammation networks in tumor-stroma interaction in breast cancer [106]. Upregulation of Notch signaling in TNBC leads to cytokine secretion, which recruits macrophages in the tumor stroma, with a role for USP9X upstream of Notch [107]. Single cell transcriptomic analysis revealed that Jagged/Notch signaling regulates immune cell homeostasis, and treatment with an anti-Jagged1 antibody delayed tumor recurrence in a mouse model [108]. Conversely, myeloid-derived suppressor cells (MDSCs) can induce Notch signaling in a breast cancer model [109], while dysregulated Notch signaling induces MDSCs in a mouse model for T-ALL [110].

Immune checkpoint inhibition (ICI) is a promising cancer treatment strategy, which also has emerging links to Notch signaling. ICI is rooted in the modulation of the immune response in such a way that immune targeting of cancer cells, which normally is downregulated by the tumor (see above), is unleashed, notably through activation of cytotoxic T-cells [111]. As a recognition of the importance of ICI, James Allison and Tasuku Honjo were awarded the Nobel Prize in Physiology or Medicine in 2018 for their work around two immune checkpoints: CTLA4 and PD-1/PD-L1. While this is revolutionary for certain tumors and patient groups, it should, however, be noted that ICI is not yet successful for all types of cancer, and for several cancer forms, such as lung cancer and TNBC, only a subset of immunocompetent patients benefits from the therapy [111]. Notch signaling is an emerging player in immune checkpoint regulation and, thus, it is important for ICI-based therapy development (Figure 3). An NSCLC patient cohort receiving ICI was screened for Notch mutational status, and patients with a high *NOTCH* mutation load showed enhanced immunogenicity and improved progression-free survival [112]. Similarly, NSCLC patients with *NOTCH4* mutations had a better response rate to ICI and better overall survival [113], and mutations in the Notch pathway, as well as in DNA damage response, coincided with durable clinical benefit in NSCLC [114,115]. A correlation between *NOTCH1*, *2* and *3* mutations in NSCLC and improved ICI outcome was observed [116]. Elevated Notch signaling predicts clinical benefit for ICI as well as SCLC [117], which is in line with the fact that up to 25% of SCLC patients carry inactivating mutations in *NOTCH1-4* [83] (see above). High expression of JAGGED1 in tumors correlates with macrophage infiltration and reduced T-cell response and a combined ICI PD-L1 antibody and blockade of Notch signaling reduced tumor growth in patients with triple-negative breast cancer [118]. Elevated Notch signaling in breast cancer tumor cells leads to secretion of IL1beta and CCL2 important for recruitment of tumor macrophages [119]. This also resulted in elevated TGFβ levels, which is a driver of tumor microenvironment modulation [119]. A Notch-Myc-EZH2 axis controls PD-L1 expression in chronic lymphocytic leukemia [120]. High *NOTCH3* expression correlates with increased immune checkpoint inhibitor expression and lower infiltration of CD8+ T cells in gastric cancer [121]. In conclusion, these data indicate that the level of Notch signaling may be linked to ICI response and that Notch mutational status may serve as a response biomarker for ICI in certain tumor types. 

### 3.2. Notch and Cancer-Associated Fibroblasts

Tumors have the potential to influence cells to convert into becoming cancer-associated fibroblasts (CAFs), which produce an extensive extracellular matrix in the tumor microenvironment and in tumors such as carcinomas can make up a considerable volume of the tumor [122]. Regular fibroblasts have been proposed as one important source of cells of origin for CAFs, but other cell types may also contribute [123]. We still know relatively little about the role of Notch in CAF biology, but it is an emerging view that Notch signaling plays a role in CAF activation and in the cross-talk between CAFs and the genetically mutated cells in the tumor (Figure 3). A direct Notch-mediated cross-talk between tumor cells and fibroblasts has been observed in breast ductal carcinoma in situ (DCIS). A role for Notch together with EGFR/ErbB2 signaling was originally proposed for DCIS [124,125], and a direct cross-talk between JAGGED1 on the epithelial tumor cells and NOTCH2 receptors on peritumoral fibroblasts in DCIS was later unveiled [126]. Stromal cells have also been found to elevate Notch3 signaling in breast cancer cells [127], for review see [128]. 

In lung cancer, low miR-200 expression induces Notch activation in CAFs and correlates with poor prognosis [129], corroborating an earlier study proposing a link between miR-200 and lung cancer [130]. Increased autophagy in the tumor microenvironment has been proposed to lead to downregulation of CSL and thus reduced Notch signaling, for example in CAFs from patients with squamous cell carcinoma [131]. It has, however, also been demonstrated that amplification of *NOTCH1* leads to accumulation of CAFs in the skin [132]. Notch signaling has been implicated in remodeling of the extracellular matrix in the tumor microenvironment. MMP2 and MMP4, two matrix-degrading enzymes, are regulated by Notch via NF-kB signaling [133]. Similarly, urokinase PA, which is an activator of a receptor regulating conversion of plasminogen to plasmid that leads to extracellular matrix degradation, may also be directly regulated by Notch [134]. Chronic lymphocytic leukemia (CLL) induces NOTCH2 signaling in mesenchymal stromal cells, which activates C1a, which, in turn, leads to β-activin degradation in the CLL cells [135]. In acute myeloid leukemia (AML), interactions with mesenchymal stromal cells are indispensable and activation of Notch in mesenchymal stromal cells led to enhanced AML cell proliferation [136]. 

### 3.3. Notch and the Tumor Vasculature

Tumors are capable of attracting blood vessels for their sustained growth. The resulting tumor vasculature is to some extent different compared to the normal vasculature, notably with a defective endothelium and more sparse mural cell coating, leading to higher permeability, which may be advantageous for the tumors [137]. Notch signaling is key regulator of normal vascular development. The roles of Notch in normal vascular biology range from serving a critical role in the first phase of vascular development, i.e., vasculogenesis, during which loss of Notch signaling leads to an aberrant vascular bed and embryonic death in mouse models with perturbed Notch signaling; for review, see [138]. Notch signaling is furthermore important for establishing the arterial and venous sides of the endothelium, as well as for sprouting of the blood vessels during angiogenesis. During angiogenic sprouting, Notch signaling regulates the balance between tip and stalk cells through a complex interplay with VEGF signaling, and a high level of Notch signaling leads to fewer tip cells, reduced sprouting and thus a less arborized vasculature. Conversely, low levels of Notch signaling instead promote tip cells and sprouting [139]. The Notch activity in tip-stalk decisions is largely driven by Notch1/Dll4, and with Jagged1 playing a negative regulatory role. Notch signaling regulates not only endothelial cells but also the homeostasis of mural cells, i.e., vascular smooth muscle cells (VSMC) and pericytes. Notch3 plays a key role in maintaining VSMC and loss of Notch3 signaling leads to a gradual loss of VSMC [140,141]. In keeping with this observation, the NOTCH3 gene is also mutated in the stroke and dementia syndrome Cerebral Autosomal Dominant Arteriopathy with Subcortical Infarcts and Leukoencephalopathy (CADASIL), which affects brain VSMC [142]. 

Attempts have been made to develop Notch-perturbing strategies that affect the tumor vasculature. Both vasculature-inhibiting and -promoting strategies based on modulation of Notch signaling in the tumor vasculature are being pursued, and there are currently more than 70 clinical trials in this area [143]. One strategy is based on the use of a combined NOTCH2/NOTCH3 blockade (tarextumab, OMP-59R5), which was demonstrated to affect tumor vasculature in the stroma [144]. Another approach is based on blockade of DLL4, using antibodies or Dll4-Fc moieties, which leads to a more arborized vasculature, paradoxically providing less effective vascularization with poor tumor perfusion [145,146,147]. However, it should be noted that long-term Dll4 inhibition in mice resulted in vascular neoplasms [148], suggesting that this mode of therapy cannot immediately be converted into clinical use, although there are clinical trials with Demcizumab, a humanized Dll4 antibody, which appears to be tolerated although with some side effects, including hypertension [149]. A recent interesting approach is the use of a bivalent antibody (navicixizumab) to simultaneously block DLL4 and VEGF, which shows promising results in a phase 1 clinical trial [150]. Interestingly, overexpression of Dll4 also affects tumor vasculature, leading to reduced tumor growth [151], supporting the notion of the importance of dosage sensitivity in the Notch pathway. A third strategy is based on Notch1 decoys specific for either Dll4 or Jagged1 interactions with Notch1. Here, the use of a Jagged1-inhibiting decoy led to reduced sprouting while a Dll4-inhibiting decoy promotes sprouting [152], in line with the roles of Dll4 and Jagged1 in normal vascular differentiation. 

## 4. Conclusions

Notch signaling is an evolutionarily highly conserved signaling mechanism, which is well established as an important driver for tumor initiation and progression. Mutations in the genes of the Notch pathway have been identified in several tumor forms, and the mutations can be both of the oncogenic and tumor suppressor type, depending on the tumor context. This classical view of Notch as an oncogene or tumor suppressor is however increasingly complemented by insights that Notch signaling plays roles also in the interaction between the mutated cells in the tumor and the surrounding tumor microenvironment. Progress in decoding the role of Notch in tumor-stroma interaction is made on a number of frontiers, and in this review, we have highlighted three such areas—tumor immune cells, cancer-associated fibroblasts and the tumor vasculature. It is evident that Notch signaling in various ways regulates the complex cross-talk between the tumor cells and the stromal cells, and to gain detailed insights into the molecular underpinnings of Notch signaling in these settings will be important not only in better understanding the tumorigenic process per se, but eventually also in developing Notch-based cancer therapies. 

## Figures and Tables

**Figure 1 ijms-23-06241-f001:**
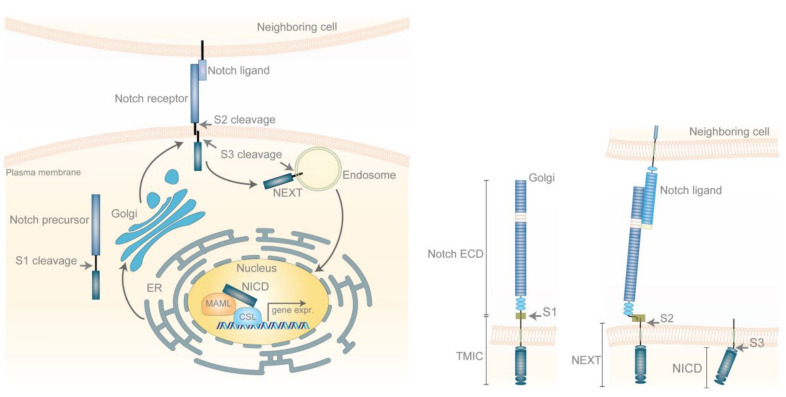
The Notch signaling pathway. To the left, a schematic representation of the routing of the Notch receptor in the signal-receiving cell is depicted. The Notch receptor is produced in the ER and undergoes S1-cleavage in the Golgi compartment. Following interaction with Notch ligand (Jagged or Dll ligand) presented on the neighboring, signal-sending cell, the Notch receptor undergoes S2-cleavage. S3-cleavage is then executed at the plasma membrane or in endosomes, and the Notch ICD is released and transported to the cell nucleus. In the cell nucleus, Notch ICD forms a ternary complex with the DNA-binding protein CSL and MAML, and the ternary complex regulates expression of specific genes, such as *Hes* or *Hey*. To the right, the various processing steps of the Notch receptor are described. S1-cleavage generates a Notch extracellular domain (Notch ECD), which is shedded, and a transmembrane and intracellular domain (TMIC) moiety. TMIC then undergoes S2-cleavage to generate the shorter Notch extracellular truncation (NEXT) moiety, which still spans the membrane. NEXT is finally cleaved by the γ-secretase complex (S3-cleavage), generating Notch ICD, which translocates to the nucleus.

**Figure 2 ijms-23-06241-f002:**
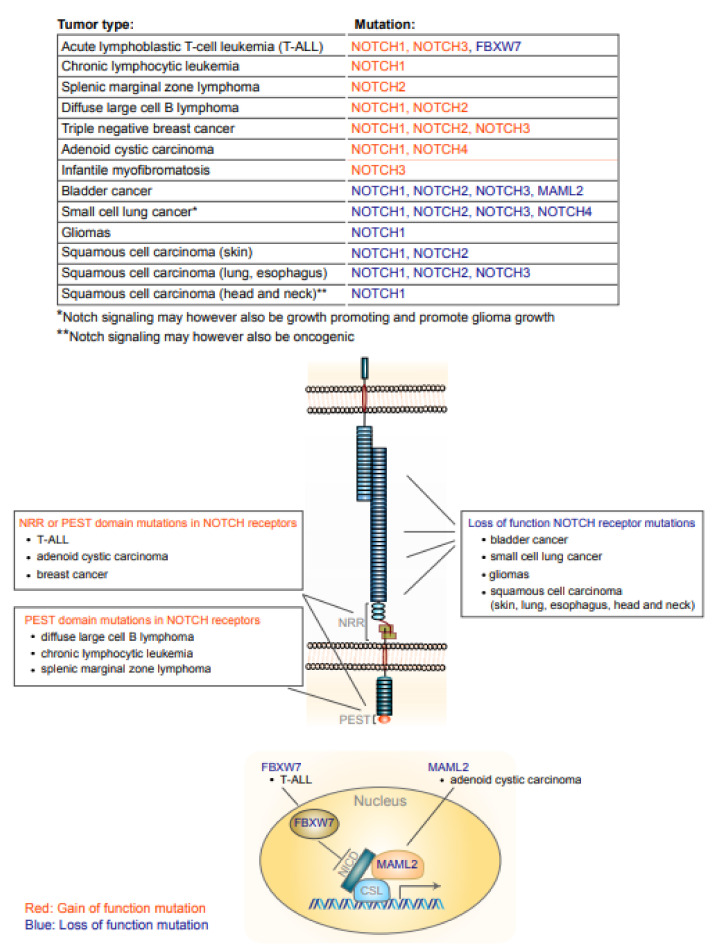
Mutations in genes of the Notch signaling pathway in different types of tumors. At the top of the figure, gain-of-function (blue) and loss-of-function (red) mutations for various types of tumors are shown. Below, a schematic representation of the Notch signaling pathway is shown, with the locations of the gain- and loss-of-function Notch mutations in the Notch receptor depicted, along with mutations in the *MAML* and *FBXW7* genes.

**Figure 3 ijms-23-06241-f003:**
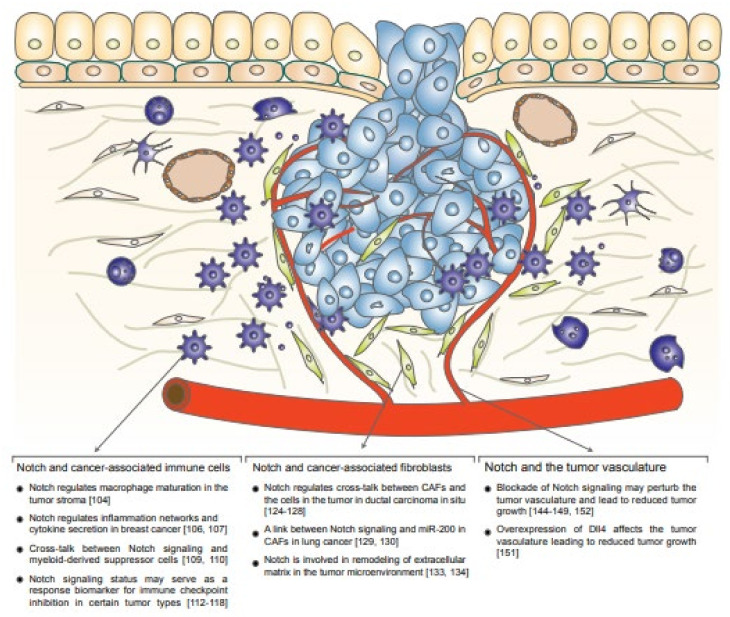
Notch and the tumor microenvironment. A schematic depiction of a tumor with its surrounding stroma. Three cell types are highlighted: cancer-associated fibroblasts, cancer-associated immune cells in the stroma and cells of the tumor vasculature. Below, examples of interactions between Notch signaling and the tumor microenvironment are provided. Numbers in brackets refer to references in the reference list.

## Data Availability

Not applicable.

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
