# Peer review of "Roles of Notch Signaling in the Tumor Microenvironment"

_ijms, 2022, doi:10.3390/ijms23116241_

Round 1
Reviewer 1 Report
Dear Authors,
this is a well-written and comprehensive review about Notch signaling. However, the complete lack of any summarizing figure or a figure showing the conception of Notch signaling or a figure mentioning mutations in components in the notch pathway and the consequenes or information about clinical perturbation interventions or downstream targets of Notch signaling.Is there a cancer-specific role of Notch signaling?
Searching for "Notch" at clinicaltrials.gov results in 393 clinical studies. Although providing much information this review has a very general character and I would strongly recomment to add a special section such as a chapter discussing clinically used inhibitors.
Author Response
Dear Authors,
this is a well-written and comprehensive review about Notch signaling.
We thank the reviewer for these positive remarks and for fair and constructive critique for how to improve the manuscript. Below we provide a point-by-point response to the comments from the reviewers and all changes in the manuscript text are marked with track change in the word file.
However, the complete lack of any summarizing figure or a figure showing the conception of Notch signaling or a figure mentioning mutations in components in the notch pathway and the consequenes or information about clinical perturbation interventions or downstream targets of Notch signaling.Is there a cancer-specific role of Notch signaling?
We appreciate this comment, and in response, we have converted Table 1, which was just a compilation of which genes in the Notch pathway that were mutated in cancer, to a full-fledged new Figure 2. In the new figure, we provide a richer depiction of the role of Notch in cancer: Specifically, we describe whether mutations are gain-of-function (increasing Notch signaling) or loss-of-function (reducing Notch signaling), to the extent that it is known for the various mutations in specific cancer forms. Furthermore, we point to where in a generic Notch receptor, the different mutations reside, as this is informative as to their gain- or loss-of-function nature. Thirdly, we also in more detail pinpoint the effect of mutations in FBXW7, an E3 ubiquitin ligase affecting Notch ICD stability and on MAML, a protein in the ternary Notch activation complex. For the other Figures (Figure 1 and former Figure 2 (now Figure 3), we believe they are informative as they stand. In Figure 1, we believe we include all processing and maturation steps of the Notch receptor, without making the figure unnecessarily complicated. For Figure 3, we believe it provides a useful “at a glance” view of the role of dysregulated Notch signaling in the tumor microenvironment and in tumor-stroma crosstalk. We also believe it nicely portrays the different cell types we address in the review (cancer immune cells, cancer-associated fibroblasts and the tumor vasculature).
Searching for "Notch" at clinicaltrials.gov results in 393 clinical studies. Although providing much information this review has a very general character and I would strongly recomment to add a special section such as a chapter discussing clinically used inhibitors.
We understand the reviewer’s point, and we in fact initially shared this idea. However, when we in the pre-submission process discussed the content and scope of the review with the Guest Editor for the “Cellular Crosstalk in the Tumor Microenvironment” special issue (Dr. Monica Ehnman, Karolinska Institute) and the IJMS Editor Jeffrey Zhu, we were strongly advised against including a clinical section and a deeper discussion on inhibitors, because the journal’s focus is more molecular and mechanistic, and not clinical. To underscore this, I attach a part of the email conversation with Dr. Zhu and his response to this question (Feb 20, 2022): “It is okay to include recent therapy development together with biology in our special issue. We mainly focus on molecular aspect, but not all of them. However, if the submission main focus on the clinical therapy development and lack of molecular aspect, our office may reject it”. Therefore, to not run the risk of being desk rejected, we tried to strike a balance, and while we indeed talk about different inhibitor approaches (both antibody and small molecule-based; see page 5 and 9), we stayed clear of going too deep into the subject, but instead focusing on the underpinning molecular mechanisms. With this said, we however appreciate the reviewer’s comment and we have therefore added a short description that there are an impressive number of clinical ongoing trials, as the reviewer correctly points out. We have also slightly elaborated on the current inhibitor research, including adding two additional references (Moellering et al., Nature 2009; Yu et al., Nature Chem Biol, 2015) (page 5, last paragraph). We leave it to the editor’s discretion whether a separate paragraph on inhibitors and clinical trials should be included.
Reviewer 2 Report
Well-written, easy to read, comprehensive review of the involvement of Notch signaling in cancer in general, and the tumor microenvironment in particular, which is based on the bibliographic analysis of up-to-date citations. Just have a couple of minor comments:
(1) The authors should make sure to use internationally accepted rules for the utilization of nuclei acid and protein.polypeptide symbols. In this regard, the italics font should be used always when referred to genes, transcripts or any other DNA or RNA forms, reserving the normal font for protein symbols.
(2) The text should be checked for relatively few instances of inappropriate use of English grammar with regard to sentence construction.
Author Response
Well-written, easy to read, comprehensive review of the involvement of Notch signaling in cancer in general, and the tumor microenvironment in particular, which is based on the bibliographic analysis of up-to-date citations. Just have a couple of minor comments:
We thank the reviewer for these positive remarks and for fair and constructive comments for how to improve the manuscript. Below we provide a point-by-point response to the comments from the reviewers and all changes in the manuscript text are marked with track change in the word file.
(1) The authors should make sure to use internationally accepted rules for the utilization of nuclei acid and protein.polypeptide symbols. In this regard, the italics font should be used always when referred to genes, transcripts or any other DNA or RNA forms, reserving the normal font for protein symbols.
We accept this critique and acknowledge that there were clear shortcomings with regard to nucleic acid/gene vs protein description. We have carefully gone through the manuscript and hopefully depicted all DNA/RNA/gene in italics, and protein in normal font in the revised version. In addition, we have double-checked the use of capital vs small case letters for human and mouse.
(2) The text should be checked for relatively few instances of inappropriate use of English grammar with regard to sentence construction.
We thank the reviewer for alerting us to some linguistic shortcomings, which have been corrected in the revised version. Here we have corrected grammatical mistakes as well as a couple of minor factual mistakes, for example around the distribution of NRR and PEST domain mutations in different tumor types, which was not clearly described in the original version.
Round 2
Reviewer 1 Report
Dear authors,
thanks for adding the requested changes.